# Unifying Graph Convolutional Neural Networks and Label Propagation

## Abstract

Label Propagation (LPA) and Graph Convolutional Neural Networks (GCN) are both message passing algorithms on graphs. Both solve the task of node classification but LPA propagates node label information across the edges of the graph, while GCN propagates and transforms node feature information. However, while conceptually similar, it is unclear how LPA and GCN can be combined under a unified framework to improve node classification. Here we study the relationship between LPA and GCN in terms of *feature/label influence*, in which we characterize how much the initial feature/label of one node influences the final feature/label of another node in GCN/LPA. Based on our theoretical analysis, we propose an end-to-end model that combines GCN and LPA. In our unified model, edge weights are learnable, and the LPA serves as regularization to assist the GCN in learning proper edge weights that lead to improved classification performance. Our model can also be seen as learning the weights for edges *based on node labels*, which is more task-oriented than existing feature-based attention models and topology-based diffusion models. In a number of experiments on real-world graphs, our model shows superiority over state-of-the-art graph neural networks in terms of node classification accuracy.

## 1 Introduction

Consider the problem of node classification in a graph, where the goal is to learn a mapping $\mathcal{M}$ : $\mathcal{V} \rightarrow \mathcal{L}$ from node set $\mathcal{V}$ to label set $\mathcal{L}$. Solution to this problem is widely applicable to various scenarios, e.g., inferring income of users in a social network or classifying scientific articles in a citation network. Different from a generic machine learning problem where samples are independent from each other, nodes are connected by edges in the graph, which provide additional information and require more delicate modeling. To capture the graph information, researchers have mainly designed models on the assumption that labels/features are correlated over the edges of the graph. In particular, on the label side $\mathcal{L}$, node labels are propagated and aggregated along edges in the graph, which is known as *Label Propagation Algorithm* (LPA) (Zhu et al., 2005; Zhou et al., 2004; Zhang & Lee, 2007; Wang & Zhang, 2008; Karasuyama & Mamitsuka, 2013; Gong et al., 2017; Liu et al., 2019a); On the node side $\mathcal{V}$, node features are propagated along edges and transformed through neural network layers, which is known as *Graph Convolutional Neural Networks* (GCN)[1] (Kipf & Welling, 2017; Hamilton et al., 2017; Li et al., 2018; Xu et al., 2018; Liao et al., 2019; Xu et al., 2019b; Qu et al., 2019).

GCN and LPA are related in that they propagate features and labels on the two sides of the mapping $\mathcal{M}$, respectively. Prior work Li et al. (2019) has shown the relationship between GCN and LPA in terms of low-pass graph filtering. However, it is unclear how the discovered relationship benefits node classification. Specifically, can GCN and LPA be combined to develop a more accurate model for node classification in graphs?

Here we study the theoretical relationship between GCN and LPA from the viewpoint of *feature/label influence*, where we quantify how much the initial feature/label of node $v_b$ influences

---

[1]There are methods in statistical relational learning Rossi et al. (2012) also using feature propagation/diffusion techniques. In this work, we focus on GCN, but the analysis and the proposed model can be easily generalized to other feature diffusion methods.

the output feature/label of node $v_a$ in GCN/LPA by studying the Jacobian/gradient of node $v_b$ with respect to node $v_a$. We also prove the quantitative relationship between feature influence and label influence, i.e., the label influence of $v_b$ on $v_a$ equals the cumulative discounted feature influence of $v_b$ on $v_a$ in expectation (Theorem 1).

Based on the theoretical analysis, we propose a unified model GCN-LPA for node classification. We show that the key to improving the performance of GCN is to enable nodes of the same class to connect more strongly with each other by making edge weights/strengths trainable. Then we prove that increasing the strength of edges between the nodes of the same class is equivalent to increasing the accuracy of LPA's predictions (Theorem 2). Therefore, we can first learn the optimal edge weights by minimizing the loss of predictions in LPA, then plug the optimal edge weights into a GCN to learn node representations. In GCN-LPA, we further combine the above two steps together and train the whole model in an end-to-end fashion, where the LPA part serves as regularization to assist the GCN part in learning proper edge weights that benefit the separation of different node classes. It is worth noticing that GCN-LPA can also be seen as learning the weights for edges based on *node label information*, which requires less handcrafting and is more task-oriented than existing attention models that learn edge weights based on *node feature similarity* (Veličković et al., 2018; Thekumparampil et al., 2018; Zhang et al., 2018; Liu et al., 2019b) or diffusion models that learn adjacency matrix based on *graph topology* (Klicpera et al., 2019a; Xu et al., 2019a; Abu-El-Haija et al., 2019; Klicpera et al., 2019b).

We conduct extensive experiments on five datasets, and the results indicate that our model outperforms state-of-the-art graph neural networks in terms of classification accuracy. The experimental results also show that combining GCN and LPA together is able to learn more informative edge weights thereby leading to better performance.

## 2 OUR APPROACH

In this section, we first formulate the node classification problem and briefly introduce LPA and GCN. We then prove their relationship from the viewpoints of feature influence and label influence. Based on the theoretical finding, we propose a unified model GCN-LPA, and analyze why our model is theoretically superior to vanilla GCN.

### 2.1 PROBLEM FORMULATION AND PRELIMINARIES

Consider a graph $\mathcal{G} = (\mathcal{V}, A, X, Y)$, where $\mathcal{V} = \{v_1, \cdots, v_n\}$ is the set of nodes, $A \in \mathbb{R}^{n \times n}$ is the adjacency matrix, $X$ is the feature matrix of nodes and $Y$ is labels of nodes. $a_{ij}$ (the $ij$-th entry of $A$) is the weight of the edge connecting $v_i$ and $v_j$. $\mathcal{N}(v)$ denotes the set of first-order neighbors of node $v$ in graph $\mathcal{G}$. Each node $v_i$ has a feature vector $\mathbf{x}_i$ which is the $i$-th row of $X$, while only the first $m$ nodes ($m \ll n$) have labels $y_1, \cdots, y_m$ from a label set $\mathcal{L} = \{1, \cdots, c\}$. The goal is to learn a mapping $\mathcal{M} : \mathcal{V} \to \mathcal{L}$ and predict labels of unlabeled nodes.

**Label Propagation Algorithm**. LPA (Zhu et al., 2005) assumes that two connected nodes are likely to have the same label, and thus it propagates labels iteratively along the edges. Let $Y^{(k)} = [y_1^{(k)}, \cdots, y_n^{(k)}]^\top \in \mathbb{R}^{n \times c}$ be the soft label matrix in iteration $k > 0$, in which the $i$-th row $y_i^{(k)\top}$ denotes the predicted label distribution for node $v_i$ in iteration $k$. When $k = 0$, the initial label matrix $Y^{(0)} = [y_1^{(0)}, \cdots, y_n^{(0)}]^\top$ consists of one-hot label indicator vectors $y_i^{(0)}$ for $i = 1, \cdots, m$ (i.e., labeled nodes) or zero vectors otherwise (i.e., unlabeled nodes). Then LPA in iteration $k$ is formulated as the following two steps:

$$Y^{(k+1)} = \tilde{A} Y^{(k)}, \tag{1}$$

$$y_i^{(k+1)} = y_i^{(0)}, \ \forall \, i \le m. \tag{2}$$

In the above equations, $\tilde{A}$ is the normalized adjacency matrix, which can be the random walk transition matrix $\tilde{A}_{rw} = D^{-1}A$ or the symmetric transition matrix $\tilde{A}_{sym} = D^{-\frac{1}{2}}AD^{-\frac{1}{2}}$, where $D$ is the diagonal degree matrix for $A$ with entries $d_{ii} = \sum_j a_{ij}$. Without loss of generosity, we use $\tilde{A} = \tilde{A}_{rw}$ in this work. In Eq. (1), all nodes propagate labels to their neighbors according to normalized edge weights. Then in Eq. (2), labels of all labeled nodes are reset to their initial values,

because LPA wants to persist labels of nodes which are labeled, so that unlabeled nodes do not overpower the labeled ones as the initial labels would otherwise fade away.

**Graph Convolutional Neural Networks**. GCN Kipf & Welling (2017) is a multi-layer feedforward neural network that propagates and transforms node features across the graph. The feature propagation scheme of GCN in layer $k$ is:

$$X^{(k+1)} = \sigma\left(\tilde{A} X^{(k)} W^{(k)}\right), \tag{3}$$

where $W^{(k)}$ is trainable weight matrix in the $k$-th layer, $\sigma(\cdot)$ is an activation function, and $X^{(k)} = [\mathbf{x}_1^{(k)}, \cdots, \mathbf{x}_n^{(k)}]^\top$ are the $k$-th layer node representations with $X^{(0)} = X$. By setting the dimension of the last layer to the number of classes $c$, the last layer can be seen as (unnormalized) label distribution predicted for a given node. The whole model can thus be optimized by minimizing the discrepancy between predicted node label distributions and ground-truth labels $Y$.

## 2.2 FEATURE INFLUENCE AND LABEL INFLUENCE

Consider two nodes $v_a$ and $v_b$ in a graph. Inspired by Koh & Liang (2017) and Xu et al. (2018), we study the relationship between GCN and LPA in terms of influence, i.e., how the output feature/label of $v_a$ will change if the initial feature/label of $v_b$ is varied slightly. Technically, the feature/label influence is measured by the Jacobian/gradient of the output feature/label of $v_a$ with respect to the initial feature/label of $v_b$. Denote $\mathbf{x}_a^{(k)}$ as the $k$-th layer representation vector of $v_a$ in GCN, and $\mathbf{x}_b$ as the initial feature vector of $v_b$. We quantify the feature influence of $v_b$ on $v_a$ as follows:

**Definition 1 (Feature influence)** *The feature influence of node $v_b$ on node $v_a$ after $k$ layers of GCN is the L1-norm of the expected Jacobian matrix $\partial \mathbf{x}_a^{(k)}/\partial \mathbf{x}_b$: $I_f(v_a, v_b; k) = \left\|\mathbb{E}\left[\partial \mathbf{x}_a^{(k)}/\partial \mathbf{x}_b\right]\right\|_1$. The normalized feature influence is then defined as $\tilde{I}_f(v_a, v_b; k) = I_f(v_a, v_b; k)/\sum_{v_i \in \mathcal{V}} I_f(v_a, v_i; k)$.*

We also consider the label influence of node $v_b$ on node $v_a$ in LPA (this implies that $v_a$ is unlabeled and $v_b$ is labeled). Since different label dimensions of $y_i^{(\cdot)}$ do not interact with each other in LPA, we assume that all $y_i$ and $y_i^{(\cdot)}$ are scalars within $[0, 1]$ (i.e., this is a binary classification task) for simplicity. Label influence is defined as follows:

**Definition 2 (Label influence)** *The label influence of labeled node $v_b$ on unlabeled node $v_a$ after $k$ iterations of LPA is the gradient of $y_a^{(k)}$ with respect to $y_b$: $I_l(v_a, v_b; k) = \partial y_a^{(k)}/\partial y_b$.*

The following theorem shows the relationship between feature influence and label influence:

**Theorem 1 (Relationship between feature influence and label influence)** *Assume the activation function used in GCN is ReLU. Denote $v_a$ as an unlabeled node, $v_b$ as a labeled node, and $\beta$ as the fraction of unlabeled nodes. Then the label influence of $v_b$ on $v_a$ after $k$ iterations of LPA equals, in expectation, to the cumulative normalized feature influence of $v_b$ on $v_a$ after $k$ layers of GCN:*

$$\mathbb{E}\left[I_l(v_a, v_b; k)\right] = \sum_{j=1}^k \beta^j \tilde{I}_f(v_a, v_b; j). \tag{4}$$

Proof of Theorem 1 is in Appendix A. Intuitively, Theorem 1 shows that if $v_b$ has high label influence on $v_a$, then the initial feature vector of $v_b$ will also affect the output feature vector of $v_a$ greatly. Theorem 1 provides the theoretical guideline for designing our unified model in the next subsection.

## 2.3 THE UNIFIED MODEL

Before introducing the proposed model, we rethink the GCN method and see what an ideal set of node representations should be like. Since we aim to classify nodes, the perfect node representation would be such that nodes with the same label are embedded closely together, which would give a large separation between different classes. Intuitively, the key to achieve this goal is to enable nodes within the same class to connect more strongly with each other, so that they are pushed together by GCN (more discussion is presented in Section 2.4). We can therefore make edge strengths/weights

trainable, then learn to increase the *intra-class feature influence*: $\sum_{i \in \mathcal{L}} \sum_{v_a, v_b: y_a = i, y_b = i} \tilde{I}_f(v_a, v_b)$ ($\mathcal{L}$ is the label set), by adjusting edge weights. However, this requires operating on Jacobian matrices with the size of $d^{(0)} \times d^{(K)}$ ($d^{(0)}$ and $d^{(K)}$ are the dimensions of input and output in GCN, respectively), which is impractical if initial node features are high-dimensional. Fortunately, we can turn to optimizing the *intra-class label influence* instead, i.e., $\sum_{i \in \mathcal{L}} \sum_{v_a, v_b: y_a = i, y_b = i} I_l(v_a, v_b)$, according to Theorem 1. Note that $\sum_{i \in \mathcal{L}} \sum_{v_a, v_b: y_a = i, y_b = i} I_l(v_a, v_b) = \sum_{v_a} \sum_{v_b: y_b = y_a} I_l(v_a, v_b)$. We further show, by the following theorem, that the term $\sum_{v_b: y_b = y_a} I_l(v_a, v_b)$ (the total intra-class label influence on a given node $v_a$) is proportional to the probability that $v_a$ is classified correctly by LPA:

**Theorem 2 (Relationship between label influence and LPA's prediction)** *Consider a given node $v_a$ and its label $y_a$. If we treat node $v_a$ as unlabeled, then the total label influence of nodes with label $y_a$ on node $v_a$ is proportional to the probability that node $v_a$ is classified as $y_a$ by LPA:*

$$\sum_{v_b: y_b = y_a} I_l(v_a, v_b; k) \propto \Pr\left(\hat{y}_a^{lpa} = y_a\right), \tag{5}$$

*where $\hat{y}_a^{lpa}$ is the predicted label of $v_a$ using a $k$-iteration LPA.*

Proof of Theorem 2 is in Appendix B. Theorem 2 indicates that, if edge weights $\{a_{ij}\}$ maximize the probability that $v_a$ is correctly classified by LPA, then they also maximize the intra-class label influence on node $v_a$. We can therefore first learn the optimal edge weights $A^*$ by minimizing the loss of predicted labels by LPA:[2]

$$A^* = \arg\min_A L_{lpa}(A) = \arg\min_A \frac{1}{m} \sum_{v_a: a \leq m} J(\hat{y}_a^{lpa}, y_a), \tag{6}$$

where $J$ is the cross-entropy loss, $\hat{y}_a^{lpa}$ and $y_a$ are the predicted label distribution of $v_a$ using LPA and the true one-hot label of $v_a$, respectively. $a \leq m$ means $v_a$ is labeled. The optimal $A^*$ maximizes the probability that each node is correctly labeled by LPA, thus also maximizes the intra-class label influence (according to Theorem 2) and intra-class feature influence (according to Theorem 1). Since $A^*$ increases the connection strength within each class, it is expected to improve the performance of GCN compared with the original adjacency matrix $A$. Therefore, we can plug $A^*$ into GCN to predict labels:

$$X^{(k+1)} = \sigma(A^* X^{(k)} W^{(k)}), \; k = 0, 1, \cdots, K - 1. \tag{7}$$

We use $\hat{y}_a^{gcn}$, the $a$-th row of $X^{(K)}$, to denote the predicted label distribution of $v_a$ using the GCN specified in Eq. (7). Then the optimal transformation matrices in the GCN can be learned by minimizing the loss of predicted labels by GCN:

$$W^* = \arg\min_W L_{gcn}(W, A^*) = \arg\min_W \frac{1}{m} \sum_{v_a: a \leq m} J(\hat{y}_a^{gcn}, y_a), \tag{8}$$

It is more elegant (and empirically better) to combine the above two steps together into a multi-objective optimization problem, and train the whole model in an end-to-end fashion:

$$W^*, A^* = \arg\min_{W, A} L_{gcn}(W, A) + \lambda L_{lpa}(A), \tag{9}$$

where $\lambda$ is the balancing hyper-parameter. In this way, $L_{lpa}(A)$ serves as a regularization term that assists the learning of edge weights $A$, since it is hard for GCN to learn both $W$ and $A$ simultaneously due to overfitting. The proposed GCN-LPA approach can also be seen as learning the importance of edges that can be used to reconstruct node labels accurately by LPA, then transferring this knowledge from label space to feature space for GCN.

It is also worth noticing how the optimal $A^*$ is configured. The principle here is that we do not modify the basic structure of the original graph (i.e., not adding or removing edges) but only adjusting weights of existing edges. This is equivalent to learning a positive mask matrix $M$ for the adjacency matrix $A$ and taking the Hadamard product $M \circ A = A^*$. Each element $M_{ij}$ can be set as either a free variable or a function of the two nodes, for example, $M_{ij} = \log\left(\exp(\mathbf{x}_i^\top \mathbf{H} \mathbf{x}_j) + 1\right)$ where $\mathbf{H}$ is a learnable kernel matrix for measuring feature similarity.

---

[2]Here the optimal edge weights $A^*$ share the same topology as the original graph $\mathcal{G}$, i.e., we do not add or remove edges from $\mathcal{G}$ but only learning the weights of existing edges. See the end of this subsection for more discussion.

## 2.4 ANALYSIS OF GCN-LPA MODEL BEHAVIOR

In this subsection, we show benefits of our unified model compared with GCN by analyzing properties of embeddings produced by the two models. We first analyze the update rule of GCN for node $v_i$: $\mathbf{x}_i^{(k+1)} = \sigma\left(\sum_{v_j \in \mathcal{N}(v_i)} \tilde{a}_{ij} \mathbf{x}_j^{(k)} W^{(k)}\right)$, where $\tilde{a}_{ij} = a_{ij}/d_{ii}$ is the normalized weight of edge $(j, i)$. This formula can be decomposed into the following two steps: (1) In *aggregation* step, we calculate the aggregated representation $\mathbf{h}_i^{(k)}$ of all neighborhoods $\mathcal{N}(v_i)$: $\mathbf{h}_i^{(k)} = \sum_{v_j \in \mathcal{N}(v_i)} \tilde{a}_{ij} \mathbf{x}_j^{(k)}$. (2) In *transformation* step, the aggregated representation $\mathbf{h}_i^{(k)}$ is mapped to a new space by a transformation matrix and nonlinear function: $\mathbf{x}_i^{(k+1)} = \sigma\left(\mathbf{h}_i^{(k)} W^{(k)}\right)$. We show by the following theorem that the aggregation step reduces the overall distance in the embedding space between the nodes that are connected in the graph:

**Theorem 3 (Shrinking property in GCN)** *Let $D(\mathbf{x}) = \frac{1}{2} \sum_{v_i, v_j} \tilde{a}_{ij} \|\mathbf{x}_i - \mathbf{x}_j\|_2^2$ be a distance metric over node embeddings* $\mathbf{x}$*. Then we have $D(\mathbf{h}^{(k)}) \leq D(\mathbf{x}^{(k)})$.*

Proof of Theorem 3 is in Appendix C. Theorem 3 indicates that the overall distance among connected nodes is reduced after taking one aggregation step, which implies that connected components in the graph "shrink" and nodes within each connected component get closer to each other in the embedding space. In an ideal case where edges only connect nodes with the same label, the aggregation step will push nodes within the same class together, which greatly benefits the transformation step that acts like using a hyperplane $W^{(k)}$ for classification. However, two connected nodes may have different labels. These "noisy" edges will impede the formation of clusters and make the interclass boundary less clear.

Fortunately, in GCN-LPA, edge weights are learned by minimizing the difference between ground-truth labels and labels reconstructed

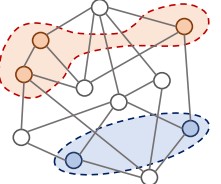 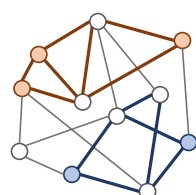

(a) A graph with two classes of nodes

(b) Potential intra-class edges (bold links)

Figure 1: A graph with two classes of nodes, while white nodes are unlabeled (Figure 1a). To classify nodes, our model will increase the connecting strength among nodes within the same class, thereby increasing their feature/label influence on each other. In this way, our model is able to identify potential intra-class edges (bold links in Figure 1b) and strengthen their weights.

from local neighbors. This will force the model to increase the weight/bandwidth of possible paths that connect nodes with the same label, so that labels can "flow" easily along these paths for the purpose of label reconstruction. In this way, GCN-LPA is able to identify potential intra-class edges and increase their weights to assist learning clustering structures ( see Figure 1 for an illustrating example).

To empirically justify our claim, we apply a two-layer untrained GCN with randomly initialized transformation matrices to the well-known Zachary's karate club network (Zachary, 1977) as shown in Figure 2a, which contains 34 nodes of 2 classes and 78 unweighted edges (grey solid lines). We then increase the weights of intra-class edges by ten times to simulate GCN-LPA. We find that GCN works well on this network (Figure 2b), but GCN-LPA performs even better than GCN because the node embeddings are completely linearly separable as shown in Figure 2c. To further justify our claim, we randomly add 20 "noisy" inter-class edges (grey dotted lines) to the original network, from which we observe that GCN is misled by noise and mixes nodes of two classes together (Figure 2d), but GCN-LPA still distinguishes the two clusters (Figure 2e) because it is better at "denoising" undesirable edges based on the supervised signal of labels.

## 3 CONNECTION TO EXISTING WORK

Edge weights play a key role in graph-based machine learning algorithms. In this section, we discuss three lines of related work that learn edge weights adaptively.

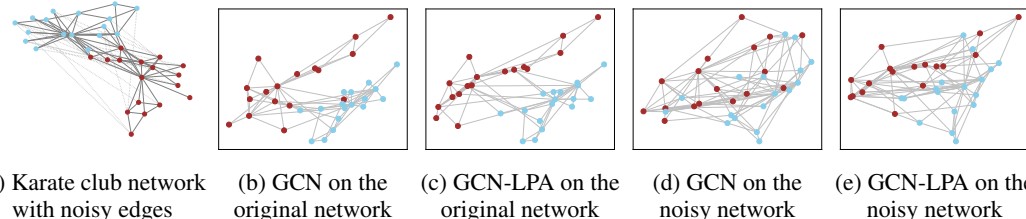

(a) Karate club network with noisy edges    (b) GCN on the original network    (c) GCN-LPA on the original network    (d) GCN on the noisy network    (e) GCN-LPA on the noisy network

Figure 2: Node embeddings of Zachary's karate club network trained on a node classification task (red vs. blue). Figure 2a visualizes the graph. Node coordinates in Figure 2b-2e are the embedding coordinates. Notice that GCN does not produce linearly separable embeddings (Figure 2b vs. Figure 2c), while GCN-LPA performs much better even in the presence of noisy edges (Figure 2d vs. Figure 2e). Additional visualizations are included in Appendix D.

**Locally Linear Embedding**. Locally linear embedding (LLE) (Roweis & Saul, 2000) and its variants (Zhang & Wang, 2007; Kong et al., 2012) learn edge weights by constructing a linear dependency between a node and its neighbors, then use the learned edge weights to embed high-dimensional nodes into a low-dimensional space. Our work is similar to LLE in the aspect of transferring the knowledge of edge importance from one space to another, but the difference is that LLE is an unsupervised dimension reduction method that learns the graph structure based on local proximity only, while our work is semi-supervised and explores high-order relationship among nodes.

**Label Propagation Algorithm**. Classical LPA (Zhu et al., 2005; Zhou et al., 2004) can only make use of node labels rather than node features. In contrast, adaptive LPA considers node features by making edge weights learnable. Typical techniques of learning edge weights include adopting kernel functions (Zhu et al., 2003; Liu et al., 2019a) (e.g., $a_{ij} = \exp(-\sum_d (x_{id} - x_{jd})^2 / \sigma_d^2)$ where $d$ is dimensionality of features), minimizing neighborhood reconstruction error (Wang & Zhang, 2008; Karasuyama & Mamitsuka, 2013), using leave-one-out loss (Zhang & Lee, 2007), or imposing sparseness on edge weights (Hong et al., 2009). However, in these LPA variants, node features are only used to assist learning the graph structure rather than explicitly mapped to node labels, which limits their capability in node classification. Another notable difference is that adaptive LPA learns edge weights by introducing the regularizations above, while our work takes LPA itself as regularization to learn edge weights.

**Attention and Diffusion on Graphs**. Our method is also conceptually connected to attention mechanism on graphs, in which an attention weight $\alpha_{ij}$ is learned between node $v_i$ and $v_j$. For example, $\alpha_{ij} = \text{LeakyReLU}(\boldsymbol{a}^\top [W\mathbf{x}_i || W\mathbf{x}_j])$ in GAT (Veličković et al., 2018), $\alpha_{ij} = a \cdot \cos(W\mathbf{x}_i, W\mathbf{x}_j)$ in AGNN (Thekumparampil et al., 2018), $\alpha_{ij} = (W_1\mathbf{x}_i)^\top W_2\mathbf{x}_j$ in GaAN (Zhang et al., 2018), and $\alpha_{ij} = \boldsymbol{a}^\top \tanh(W_1\mathbf{x}_i + W_2\mathbf{x}_j)$ in GeniePath (Liu et al., 2019b), where $a$ and $W$ are trainable variables. Our method is also similar to diffusion-based methods (Klicpera et al., 2019a; Xu et al., 2019a; Abu-El-Haija et al., 2019; Klicpera et al., 2019b; Jiang et al., 2019; Yang et al., 2019). Graph diffusion uses extended neighborhoods for aggregation in GNNs, which can be seen as learning a new adjacency matrix for a given graph. A significant difference between attention/diffusion mechanisms and our work is that attention/diffusion is learned based on feature similarity/graph topology, while we propose that edge weights should be consistent with the distribution of labels on the graph, which requires less handcrafting of the attention/diffusion function and is more task-oriented.

# 4 EXPERIMENTS

## 4.1 EXPERIMENT SETUP

**Datasets**. We use the following five datasets in our experiments. **Cora**, **Citeseer**, and **Pubmed** (Sen et al., 2008) are citation networks, where nodes correspond to documents, edges correspond to citation links, and each node has a sparse bag-of-words feature vector as well as a class label. We also use two co-authorship networks (Shchur et al., 2018), **Coauthor-CS** and **Coauthor-Phy**, where nodes are authors and an edge indicates that two authors co-authored a paper. Node features represent paper keywords for each author's papers, and class labels indicate most active fields of

| Method | Cora | Citeseer | Pubmed | Coauthor-CS | Coauthor-Phy |
|---|---|---|---|---|---|
| LR | $57.1 \pm 2.3$ | $61.0 \pm 2.2$ | $64.1 \pm 3.1$ | $86.4 \pm 0.9$ | $86.7 \pm 1.5$ |
| LPA | $74.4 \pm 2.6$ | $67.8 \pm 2.1$ | $70.5 \pm 5.3$ | $73.6 \pm 3.9$ | $86.6 \pm 2.0$ |
| GCN | $81.4 \pm 1.3$ | $71.9 \pm 1.9$ | $77.5 \pm 2.9$ | $91.1 \pm 0.5$ | $92.4 \pm 1.0$ |
| GAT | $80.7 \pm 1.3$ | $71.4 \pm 1.9$ | $76.7 \pm 2.3$ | $90.5 \pm 0.6$ | $92.2 \pm 0.9$ |
| JK-Net | $81.3 \pm 1.4$ | $70.2 \pm 1.3$ | $77.6 \pm 0.9$ | $90.3 \pm 0.4$ | $91.0 \pm 0.7$ |
| GIN | $74.5 \pm 1.5$ | $60.7 \pm 1.3$ | $73.4 \pm 1.2$ | $84.1 \pm 1.9$ | $87.3 \pm 1.7$ |
| GDC | $\textbf{83.2} \pm 0.9$ | $72.2 \pm 1.4$ | $77.8 \pm 0.8$ | $91.4 \pm 1.0$ | $92.0 \pm 0.7$ |
| GCN+LPA | $78.4 \pm 0.7$ | $69.8 \pm 1.4$ | $74.1 \pm 0.9$ | $84.5 \pm 1.0$ | $89.7 \pm 0.8$ |
| GCN-LPA | $83.0 \pm 1.4$ | $\textbf{72.6} \pm 0.9$ | $\textbf{78.4} \pm 1.5$ | $\textbf{91.9} \pm 0.9$ | $\textbf{93.4} \pm 1.6$ |

Table 1: Mean and the $95\%$ confidence intervals of test set accuracy for all methods and datasets.

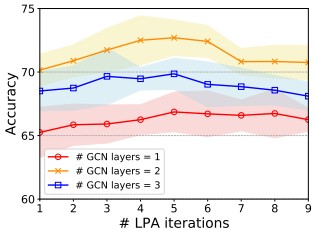

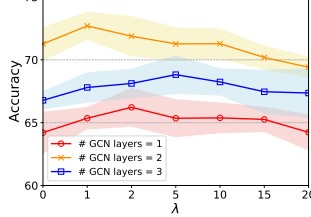

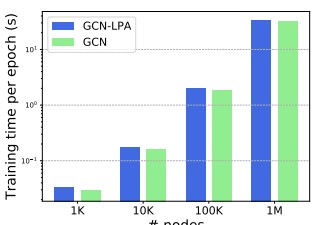

Figure 3: Sensitivity to the number of LPA iterations on Citeseer dataset.

Figure 4: Sensitivity to $\lambda$ (weight of LPA loss) on Citeseer dataset.

Figure 5: Training time per epoch on random graphs.

study for each author. Statistics of the five datasets are shown in Appendix E. We also calculate the intra-class edge rate (the fraction of edges that connect two nodes within the same class), which is significantly higher than inter-class edge rate in all networks. The finding supports our claim in Section 2.4 that node classification benefits from intra-class edges in a graph.

**Baselines**. We compare against the following baselines in our experiments. **Logistic Regression (LR)** is feature-based methods that do not consider the graph structure. **Label Propagation (LPA)** (Zhu et al., 2005), on the other hand, only consider the graph structure and ignore node features. We also compare with several GNNs: **Graph Convolutional Network (GCN)** (Kipf & Welling, 2017), **Graph Attention Network (GAT)**, **Jumping Knowledge Network (JK-Net)** (Xu et al., 2018), **Graph Isomorphism Network (GIN)** (Xu et al., 2019b), and **Graph Diffusion Convolution (GDC)** (Klicpera et al., 2019b) (with GCN as the base model). In addition, we propose another baseline **GCN+LPA**, which simply adds predictions of GCN and LPA together.

**Experimental Setup**. Our experiments focus on the transductive setting where we only know labels of part of nodes but have access to the entire graph as well as features of all nodes.[3] We randomly sample 20 nodes per class as training set, 50 nodes per class as validation set, and the remaining nodes as test set. The weight of each edge is treated as a free variable during training. We train our model for 200 epochs using Adam (Kingma & Ba, 2015) and report the test set accuracy when validation set accuracy is maximized. Each experiment is repeated five times and we report the mean and the $95\%$ confidence interval. We initialize weights according to Glorot & Bengio (2010) and row-normalize input features. During training, we apply L2 regularization to the transformation matrices and use the dropout technique (Srivastava et al., 2014). The settings of all other hyperparameters can be found in Appendix F.

### 4.2 RESULTS

**Comparison with Baselines**. The results of node classification are summarized in Table 1. Table 1 indicates that only using node features (LR) or graph structure (LPA) will lead to information loss and cannot fully exploit datasets. The results demonstrate that our proposed GCN-LPA model surpasses state-of-the-art GNN baselines. We notice that GDC is a strong baseline on Cora, but it

---

[3]Our method can be easily generalized to inductive setting if implemented using minibatch training like GraphSAGE (Hamilton et al., 2017).

| Labeled node rate | 5% | 10% | 20% | 50% | 80% |
|---|---|---|---|---|---|
| LPA | $67.9 \pm 2.1$ | $68.1 \pm 1.3$ | $70.5 \pm 1.5$ | $72.5 \pm 1.2$ | $76.4 \pm 1.1$ |
| GCN | $72.1 \pm 1.9$ | $72.5 \pm 1.8$ | $74.3 \pm 0.9$ | $76.8 \pm 0.6$ | $80.2 \pm 1.5$ |
| GCN-LPA | $\mathbf{72.7} \pm 1.2$ | $\mathbf{73.2} \pm 1.1$ | $\mathbf{75.4} \pm 1.5$ | $\mathbf{78.2} \pm 1.3$ | $\mathbf{82.3} \pm 0.9$ |

Table 2: Accuracy of LPA, GCN, and GCN-LPA on Citeseer with different labeled node rate.

does not perform consistently well on other datasets. In addition, GCN+LPA does not perform well, since it utilizes the prediction of LPA directly, making its performance limited by LPA.

**Efficacy of LPA Regularization**. We investigate the influence of the number of LPA iterations and the training weight of LPA loss term $\lambda$ on the performance of classification. The results on Citeseer dataset are plotted in Figures 3 and 4, respectively, where each line corresponds to a given number of GCN layers in GCN-LPA. From Figure 3 we observe that the performance is boosted at first when the number of LPA iterations increases, then the accuracy stops increasing and decreases since a large number of LPA iterations will include more noisy nodes. Figure 4 shows that training without the LPA loss term (i.e., $\lambda = 0$) is more difficult than the case where $\lambda = 1 \sim 5$, which justifies our aforementioned claim that it is hard for the GCN part to learn both transformation matrices $W$ and edge weights $A$ simultaneously without the assistance of LPA regularization.

**Influence of Labeled Node Rate**. To study the influence of labeled node rate on the performance of our model, we vary the ratio of labeled node rate on Citeseer from $5\%$ to $80\%$ while keeping the validation and test set fixed, and report the result in Table 2. From Table 2 we observe that GCN-LPA outperforms GCN and LPA consistently, and the improvement achieved by GCN-LPA increases when labeled node rate is larger (from 0.6% to 2.1% compared with GCN). This is because GCN-LPA requires node labels to calculate edge weights. Therefore, a larger labeled node rate will provide more information for identifying noisy edges.

**Visualization of Learned Edge Weights**. To intuitively understand what our model learns about edge weights, we split nodes in Coauthor-CS dataset into 15 groups according to their labels, and calculate the average weights of edges connecting every pair of node groups as well as the average weights of edges within every group. The results are shown in Figure 6, where darker color indicates higher average weights of edges. It is clear that values along the diagonal (intra-class edges weights) are significantly larger than off-diagonal values (inter-class edge weights) in general, which demonstrates that GCN-LPA is able to identify the importance of edges and distinguish inter-class and intra-class edges. The visualization results are similar for other datasets.

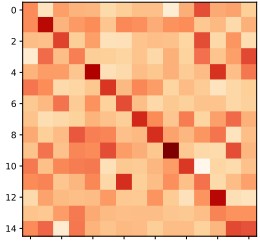

Figure 6: Visualization of learned edge weights in Coauthor-CS dataset.

**Time Complexity**. We study the training time of GCN-LPA on random graphs. We use the one-hot identity vector as feature and 0 as label for each node. The size of training set and validation set is 100 and 200, respectively, while the rest is test set. The average number of neighbors for each node is set as 5, and the number of nodes is varied from one thousand to one million. We run GCN-LPA and GCN for 100 epochs on a Microsoft Azure virtual machine with 1 NVIDIA Tesla M60 GPU, 12 Intel Xeon CPUs (E5-2690 v3 @2.60GHz), and 128GB of RAM, using the same hyper-parameter setting as in Cora. The training time per epoch of GCN-LPA and GCN is presented in Figure 5. Our result shows that GCN-LPA requires only $9.2\%$ extra training time on average compared to GCN.

## 5 CONCLUSION

We studies the theoretical relationship between two types of well-known graph-based algorithms for node classification, label propagation algorithm and graph convolutional neural networks, from the perspectives of feature/label influence. We then propose a unified model GCN-LPA, which learns transformation matrices and edge weights simultaneously in GCN with the assistance of LPA regularizer. We also analyze why our unified model performs better than traditional GCN in terms of node classification. Experiments on five datasets demonstrate that our model outperforms state-of-the-art baselines, and it is also highly time-efficient with respect to the size of a graph.

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

APPENDIX

A   PROOF OF THEOREM 1

Before proving Theorem 1, we first give two lemmas that demonstrate the exact form of feature influence and label influence defined in this paper. The relationship between feature influence and label influence can then be deduced from their exact forms.

**Lemma 1** *Assume that the nonlinear activation function in GCN is ReLU. Let $\mathcal{P}_k^{a \to b}$ be a path $[v^{(k)}, v^{(k-1)}, \cdots, v^{(0)}]$ of length $k$ from node $v_a$ to node $v_b$, where $v^{(k)} = v_a$, $v^{(0)} = v_b$, and $v^{(i-1)} \in \mathcal{N}(v^{(i)})$ for $i = k, \cdots, 1$. Then we have*

$$\tilde{I}_f(v_a, v_b; k) = \sum_{\mathcal{P}_k^{a \to b}} \prod_{i=k}^{1} \tilde{a}_{v^{(i-1)}, v^{(i)}}, \tag{10}$$

*where $\tilde{a}_{v^{(i-1)}, v^{(i)}}$ is the normalized weight of edge $(v^{(i)}, v^{(i-1)})$.*

*Proof.*   See Xu et al. (2018) for the detailed proof.                                                      □

The product term in Eq. (10) is the probability of a given path $\mathcal{P}_k^{a \to b}$. Therefore, the right hand side in Eq. (10) is the sum over probabilities of all possible paths of length $k$ from $v_a$ to $v_b$, which is the probability that a random walk starting at $v_a$ ends at $v_b$ after taking $k$ steps.

**Lemma 2** *Let $\mathcal{U}_j^{a \to b}$ be a path $[v^{(j)}, v^{(j-1)}, \cdots, v^{(0)}]$ of length $j$ from node $v_a$ to node $v_b$, where $v^{(j)} = v_a$, $v^{(0)} = v_b$, $v^{(i-1)} \in \mathcal{N}(v^{(i)})$ for $i = j, \cdots, 1$, and all nodes along the path are unlabeled except $v^{(0)}$. Then we have*

$$I_l(v_a, v_b; k) = \sum_{j=1}^{k} \sum_{\mathcal{U}_j^{a \to b}} \prod_{i=j}^{1} \tilde{a}_{v^{(i-1)}, v^{(i)}}, \tag{11}$$

*where $\tilde{a}_{v^{(i-1)}, v^{(i)}}$ is the normalized weight of edge $(v^{(i)}, v^{(i-1)})$.*

To intuitively understand this lemma, note that there are two differences between Lemma 1 and Lemma 2: (1) In Lemma 1, $\tilde{I}_f(v_a, v_b; k)$ sums over all paths from $v_a$ to $v_b$ of length $k$, but in Lemma 2, $I_l(v_a, v_b; k)$ sums over all paths from $v_a$ to $v_b$ of length no more than $k$. The is because in LPA, $v_b$'s label is reset to its initial value after each iteration, which means that the label of $v_b$ serves as a constant signal that begins propagating in the graph again and again after each iteration. (2) In Lemma 1 we consider all possible paths from $v_a$ to $v_b$, but in Lemma 2, the paths are restricted to contain unlabeled nodes only. The reason here is the same as above: Since the labels of labeled nodes are reset to their initial values after each iteration in LPA, the influence of $v_b$'s label will be absorbed in labeled nodes, and the propagation of $v_b$'s label will be cut off at these nodes. Therefore, $v_b$'s label can only flow to $v_a$ along the paths with unlabeled nodes only. See Figure 7 for an illustrating example showing the label propagation in LPA.

*Proof.*   As mentioned above, a significant difference between LPA and GCN is that all labeled nodes are reset to its original labels after each iteration in LPA. This implies that the initial label $y_b$ of node $v_b$ appears not only as $y_b^{(0)}$, but also as every $y_b^{(j)}$ for $j = 1, \cdots, k-1$. Therefore, the influence of $y_b$ on $y_a^{(k)}$ is the cumulative influence of $y_b^{(j)}$ on $y_a^{(k)}$ for $j = 0, 1, \cdots, k-1$:

$$I_l(v_a, v_b; k) = \frac{\partial y_a^{(k)}}{\partial y_b} = \sum_{j=0}^{k-1} \frac{\partial y_a^{(k)}}{\partial y_b^{(j)}}. \tag{12}$$

According to the updating rule of LPA, we have

$$\frac{\partial y_a^{(k)}}{\partial y_b^{(j)}} = \frac{\partial \sum_{v_z \in \mathcal{N}(v_a)} \tilde{a}_{az} y_z^{(k-1)}}{\partial y_b^{(j)}} = \sum_{v_z \in \mathcal{N}(v_a)} \tilde{a}_{az} \frac{\partial y_z^{(k-1)}}{\partial y_b^{(j)}}. \tag{13}$$

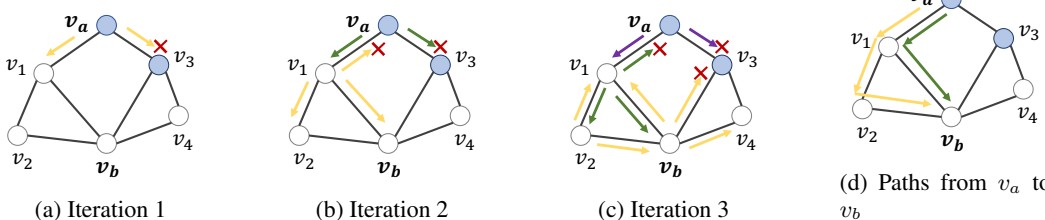

(a) Iteration 1      (b) Iteration 2      (c) Iteration 3      (d) Paths from $v_a$ to $v_b$

Figure 7: An illustrating example of label propagation in LPA. Suppose labels are propagated for three iterations, and no self-loop exists. Blue nodes are labeled while white nodes are unlabeled. (a) $v_a$'s label propagates to $v_1$ (yellow arrows). Note that the propagation of $v_a$'s label to $v_3$ is cut off since $v_3$ is labeled thus absorbing $v_a$'s label. (b) $v_a$'s label that propagated to $v_1$ further propagates to $v_2$ and $v_b$ (yellow arrows). Meanwhile, $v_a$'s label is reset to its initial value then propagates from $v_a$ again (green arrows). (c) Label propagation in iteration 3. Purple arrows denote the propagation of $v_a$'s label starting from $v_a$ for the third time. (d) All possible paths of length no more than three from $v_a$ to $v_b$ containing unlabeled nodes only. Note that there is no path of length one from $v_a$ to $v_b$.

In the above equation, the derivative $\frac{\partial y_a^{(k)}}{\partial y_b^{(j)}}$ is decomposed into the weighted average of $\frac{\partial y_z^{(k-1)}}{\partial y_b^{(j)}}$, where $v_z$ traverses all neighbors of $v_a$. For those $v_z$'s that are initially labeled, $y_z^{(k-1)}$ is reset to their initial labels in each iteration. Therefore, they are always constant and independent of $y_b^{(j)}$, meaning that their derivatives w.r.t. $y_b^{(j)}$ are zero. So we only need to consider the terms where $v_z$ is an unlabeled node:

$$\frac{\partial y_a^{(k)}}{\partial y_b^{(j)}} = \sum_{v_z \in \mathcal{N}(v_a), z>m} \tilde{a}_{az} \frac{\partial y_z^{(k-1)}}{\partial y_b^{(j)}}, \tag{14}$$

where $z > m$ means $v_z$ is unlabeled. To intuitively understand Eq. (14), one can imagine that we perform a random walk starting from node $v_a$ for one step, where the "transition probability" is the edge weights $\tilde{a}$, and all nodes in this random walk are restricted to unlabeled nodes only. Note that we can further decompose every $y_z^{(k-1)}$ in Eq. (14) in the way similar to what we do for $y_a^{(k)}$ in Eq. (13). So the expansion in Eq. (14) can be performed iteratively until the index $k$ decreases to $j$. This is equivalent to performing all possible random walks for $k-j$ steps starting from $v_a$, where all nodes but the last in the random walk are restricted to be unlabeled nodes:

$$\frac{\partial y_a^{(k)}}{\partial y_b^{(j)}} = \sum_{v_z \in \mathcal{V}} \sum_{\mathcal{U}_{k-j}^{a \to z}} \left( \prod_{i=k-j}^{1} \tilde{a}_{v^{(i-1)}, v^{(i)}} \right) \frac{\partial y_z^{(j)}}{\partial y_b^{(j)}}, \tag{15}$$

where $v_z$ in the first summation term is the end node of a random walk, $\mathcal{U}_{k-j}^{a \to z}$ in the second summation term is an unlabeled-nodes-only path from $v_a$ to $v_z$ of length $k-j$, and the product term is the probability of a given path $\mathcal{U}_{k-j}^{a \to z}$. Consider the last term $\frac{\partial y_z^{(j)}}{\partial y_b^{(j)}}$ in Eq. (15). We know that $\frac{\partial y_z^{(j)}}{\partial y_b^{(j)}} = 0$ for all $z \neq b$ and $\frac{\partial y_z^{(j)}}{\partial y_b^{(j)}} = 1$ for $z = b$, which means that only those random-walk paths that end exactly at $v_b$ (i.e., the end node $v_z$ is exactly $v_b$) count for the computation in Eq. (15). Therefore, we have

$$\frac{\partial y_a^{(k)}}{\partial y_b^{(j)}} = \sum_{\mathcal{U}_{k-j}^{a \to b}} \prod_{i=k-j}^{1} \tilde{a}_{v^{(i-1)}, v^{(i)}}, \tag{16}$$

where $\mathcal{U}_{k-j}^{a \to b}$ is a path from $v_a$ to $v_b$ of length $k-j$ containing only unlabeled nodes except $v_b$. Substituting the right hand term of Eq. (12) with Eq. (16), we obtain that

$$I_l(v_a, v_b; k) = \sum_{j=0}^{k-1} \sum_{\mathcal{U}_{k-j}^{a \to b}} \prod_{i=k-j}^{1} \tilde{a}_{v^{(i-1)}, v^{(i)}} = \sum_{j=1}^{k} \sum_{\mathcal{U}_j^{a \to b}} \prod_{i=j}^{1} \tilde{a}_{v^{(i-1)}, v^{(i)}}. \tag{17}$$

$\square$

Now Theorem 1 can be proved by combining Lemma 1 and Lemma 2:

*Proof.* Suppose that whether a node is labeled or not is independent of each other for the given graph. Then we have

$$
\begin{aligned}
\mathbb{E}\big[I_l(v_a, v_b; k)\big] &= \mathbb{E}\left[\sum_{j=1}^{k} \sum_{\mathcal{U}_j^{a \to b}} \prod_{i=j}^{1} \tilde{a}_{v^{(i-1)}, v^{(i)}}\right] = \sum_{j=1}^{k} \mathbb{E}\left[\sum_{\mathcal{U}_j^{a \to b}} \prod_{i=j}^{1} \tilde{a}_{v^{(i-1)}, v^{(i)}}\right] \\
&= \sum_{j=1}^{k} \sum_{\mathcal{P}_j^{a \to b}} \Pr\left(\mathcal{P}_j^{a \to b} \text{ is an unlabeled-nodes-only path}\right) \prod_{i=j}^{1} \tilde{a}_{v^{(i-1)}, v^{(i)}} \qquad (18) \\
&= \sum_{j=1}^{k} \sum_{\mathcal{P}_j^{a \to b}} \beta^j \prod_{i=j}^{1} \tilde{a}_{v^{(i-1)}, v^{(i)}} = \sum_{j=1}^{k} \beta^j \tilde{I}_f(v_a, v_b; j).
\end{aligned}
$$

□

## B  PROOF OF THEOREM 2

*Proof.* Denote the set of labels as $\mathcal{L}$. Since different label dimensions in $y_a^{(\cdot)}$ do not interact with each other when running LPA, the value of the $y_a$-th dimension in $y_a^{(\cdot)}$ (denoted by $y_a^{(\cdot)}[y_a]$) comes only from the nodes with initial label $y_a$. It is clear that

$$
y_a^{(k)}[y_a] = \sum_{v_b: y_b = y_a} \sum_{j=1}^{k} \sum_{\mathcal{U}_j^{a \to b}} \prod_{i=j}^{1} \tilde{a}_{v^{(i-1)}, v^{(i)}}, \qquad (19)
$$

which equals $\sum_{v_b: y_b = y_a} I_l(v_a, v_b; k)$ according to Lemma 2. Therefore, we have

$$
\Pr(\hat{y}_a = y_a) = \frac{y_a^{(k)}[y_a]}{\sum_{i \in \mathcal{L}} y_a^{(k)}[i]} \propto y_a^{(k)}[y_a] = \sum_{v_b: y_b = y_a} I_l(v_a, v_b; k) \qquad (20)
$$

□

## C  PROOF OF THEOREM 3

In this proof we assume that the dimension of node representations is one, but note that the conclusion can be easily generalized to the case of multi-dimensional representations since the function $D(\mathbf{x})$ can be decomposed into the sum of one-dimensional cases. In the following of this proof, we still use bold notations $\mathbf{x}_i^{(k)}$ and $\mathbf{h}_i^{(k)}$ to denote node representations, but keep in mind that they are scalars rather than vectors.

We give two lemmas before proving Theorem 3. The first one is about the gradient of $D(\mathbf{x})$:

**Lemma 3** $\mathbf{h}_i^{(k)} = \mathbf{x}_i^{(k)} - \frac{\partial D(\mathbf{x}^{(k)})}{\partial \mathbf{x}_i^{(k)}}$.

*Proof.* $\mathbf{x}_i^{(k)} - \frac{\partial D(\mathbf{x}^{(k)})}{\partial \mathbf{x}_i^{(k)}} = \mathbf{x}_i^{(k)} - \sum_{v_j \in \mathcal{N}(v_i)} \tilde{a}_{ij}(\mathbf{x}_i^{(k)} - \mathbf{x}_j^{(k)}) = \sum_{v_j \in \mathcal{N}(v_i)} \tilde{a}_{ij} \mathbf{x}_j^{(k)} = \mathbf{h}_i^{(k)}$. □

It is interesting to see from Lemma 3 that the aggregation step in GCN is equivalent to running gradient descent for one step with a step size of one. However, this is not able to guarantee that $D(\mathbf{h}^{(k)}) \leq D(\mathbf{x}^{(k)})$ because the step size may be too large to reduce the value of $D$.

The second lemma is about the Hessian of $D(\mathbf{x})$:

**Lemma 4** $\nabla^2 D(\mathbf{x}) \preceq 2I$, *or equivalently,* $2I - \nabla^2 D(\mathbf{x})$ *is a positive semidefinite matrix.*

*Proof.* We first calculate the Hessian of $D(\mathbf{x}) = \frac{1}{2}\sum_{v_i, v_j}\tilde{a}_{ij}\|\mathbf{x}_i - \mathbf{x}_j\|_2^2$:

$$\nabla^2 D(\mathbf{x}) = \begin{bmatrix} 1 - \tilde{a}_{11} & -\tilde{a}_{12} & \cdots & -\tilde{a}_{1n} \\ -\tilde{a}_{21} & 1 - \tilde{a}_{22} & \cdots & -\tilde{a}_{2n} \\ \vdots & \vdots & \ddots & \vdots \\ -\tilde{a}_{n1} & -\tilde{a}_{n2} & \cdots & 1 - \tilde{a}_{nn} \end{bmatrix} = I - D^{-1}A. \tag{21}$$

Therefore, $2I - \nabla^2 D(\mathbf{x}) = I + D^{-1}A$. Since $D^{-1}A$ is Markov matrix (i.e., each entry is non-negative and the sum of each row is one), its eigenvalues are within the range [-1, 1], so the eigenvalues of $I + D^{-1}A$ are within the range [0, 2]. Therefore, $I + D^{-1}A$ is a positive semidefinite matrix, and we have $\nabla^2 D(\mathbf{x}) \preceq 2I$. □

We can now prove Theorem 3:

*Proof.* Since $D$ is a quadratic function, we perform a second-order Taylor expansion of $D$ around $\mathbf{x}^{(k)}$ and obtain the following inequality:

$$\begin{aligned} D(\mathbf{h}^{(k)}) =& D(\mathbf{x}^{(k)}) + \nabla D(\mathbf{x}^{(k)})^\top(\mathbf{h}^{(k)} - \mathbf{x}^{(k)}) + \frac{1}{2}(\mathbf{h}^{(k)} - \mathbf{x}^{(k)})^\top \nabla^2 D(\mathbf{x})(\mathbf{h}^{(k)} - \mathbf{x}^{(k)}) \\ =& D(\mathbf{x}^{(k)}) - \nabla D(\mathbf{x}^{(k)})^\top \nabla D(\mathbf{x}^{(k)}) + \frac{1}{2}\nabla D(\mathbf{x}^{(k)})^\top \nabla^2 D(\mathbf{x})\nabla D(\mathbf{x}^{(k)}) \\ \leq& D(\mathbf{x}^{(k)}) - \nabla D(\mathbf{x}^{(k)})^\top \nabla D(\mathbf{x}^{(k)}) + \nabla D(\mathbf{x}^{(k)})^\top \nabla D(\mathbf{x}^{(k)}) \\ =& D(\mathbf{x}^{(k)}). \end{aligned} \tag{22}$$

□

## D MORE VISUALIZATION RESULTS ON KARATE CLUB NETWORK

Figure 8 illustrates more visualization of GCN and GCN-LPA on karate club network. In each subfigure, we vary the number of layers from 1 to 4 to examine how the learned representations evolve. The initial node features are one-hot identity vectors, and the dimension of hidden layers and output layer is 2. The transformation matrices are uniformly initialized within range [-1, 1]. We use sigmoid function as the nonlinear activation function. Comparing the four figures in each row, we conclude that the aggregation step and transformation step in GCN and GCN-LPA do benefit the separation of different classes. Comparing Figure 8a and 8c (or Figure 8b and 8d), we conclude that more inter-class edges will make the separation harder for GCN (or GCN-LPA). Comparing Figure 8a and 8b (or Figure 8c and 8d), we conclude that GCN-LPA is more noise-resistant than GCN, therefore, GCN-LPA can better differentiate classes and identify clustering substructures.

## E DATASETS DETAILS

The statistics of all datasets are shown in Table 3.

| | Cora | Citeseer | Pubmed | Coauthor-CS | Coauthor-Phy |
|---|---|---|---|---|---|
| # nodes | 2,708 | 3,327 | 19,717 | 18,333 | 34,493 |
| # edges | 5,278 | 4,552 | 44,324 | 81,894 | 247,962 |
| # features | 1,433 | 3,703 | 500 | 6,805 | 8,415 |
| # classes | 7 | 6 | 3 | 15 | 5 |
| Intra-class edge rate | 81.0% | 73.6% | 80.2% | 80.8% | 93.1% |
| Labeled node rate | 5.2% | 3.6% | 0.3% | 1.6% | 0.3% |

Table 3: Statistics for all datasets.

## F HYPER-PARAMETER SETTINGS

The detailed hyper-parameter settings for all datasets are listed in Table 4. In GCN-LPA, we use the same dimension for all hidden layers. Note that the number of GCN layers and the number of LPA iterations can actually be different since GCN and LPA are implemented as two independent modules. We use grid search to determine hyper-parameters on Cora, and perform fine-tuning on

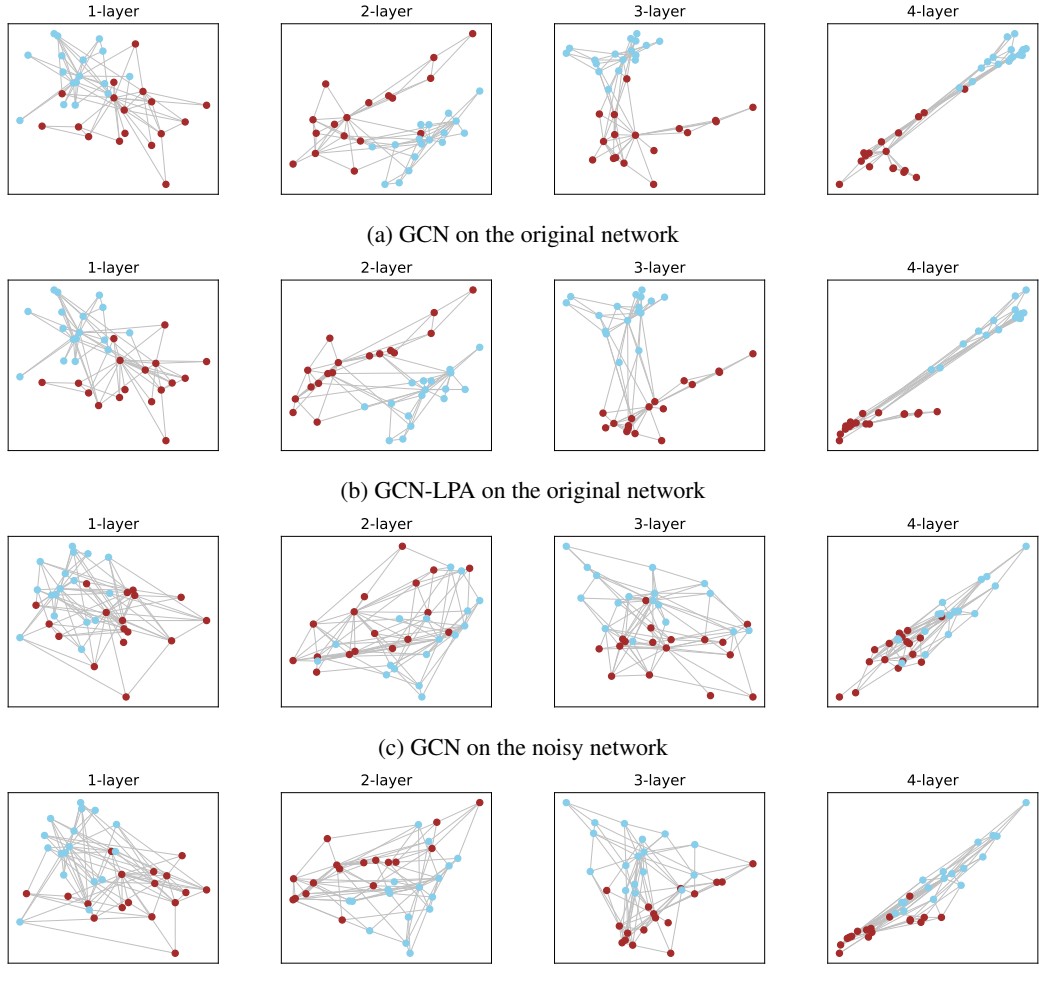

(a) GCN on the original network

(b) GCN-LPA on the original network

(c) GCN on the noisy network

(d) GCN-LPA on the noisy network

Figure 8: Visualization of GCN and GCN-LPA with $1 \sim 4$ layers on karate club network.

| | Cora | Citeseer | Pubmed | Coauthor-CS | Coauthor-Phy |
|---|---|---|---|---|---|
| Dimension of hidden layers | 32 | 16 | 32 | 32 | 32 |
| # GCN layers | 5 | 2 | 2 | 2 | 2 |
| # LPA iterations | 5 | 5 | 1 | 2 | 3 |
| L2 weight | $1 \times 10^{-4}$ | $5 \times 10^{-4}$ | $2 \times 10^{-4}$ | $1 \times 10^{-4}$ | $1 \times 10^{-4}$ |
| LPA weight ($\lambda$) | 10 | 1 | 1 | 2 | 1 |
| Dropout rate | 0.2 | 0 | 0 | 0.2 | 0.2 |
| Learning rate | 0.05 | 0.2 | 0.1 | 0.1 | 0.05 |

Table 4: Hyper-parameter settings for all datasets.

other datasets, i.e., varying one hyper-parameter per time to see if the performance can be further improved. The search spaces for hyper-parameters are as follows:

- Dimension of hidden layers: $\{8, 16, 32\}$;
- # GCN layers: $\{1, 2, 3, 4, 5, 6\}$;
- # LPA iterations: $\{1, 2, 3, 4, 5, 6, 7, 8, 9\}$;
- L2 weight: $\{10^{-7}, 2 \times 10^{-7}, 5 \times 10^{-7}, 10^{-6}, 2 \times 10^{-6}, 5 \times 10^{-6}, 10^{-5}, 2 \times 10^{-5}, 5 \times 10^{-5}, 10^{-4}, 2 \times 10^{-4}, 5 \times 10^{-4}, 10^{-3}\}$;
- LPA weight ($\lambda$): $\{0, 1, 2, 5, 10, 15, 20\}$;

- Dropout rate: $\{0, 0.1, 0.2, 0.3, 0.4, 0.5\}$;
- Learning rate: $\{0.01, 0.02, 0.05, 0.1, 0.2, 0.5\}$.

