# OpenReview forum: "Unifying Graph Convolutional Neural Networks and Label Propagation"
_ICLR.cc/2021/Conference — Reject_

### Official Review · AnonReviewer3 · 2020-10-28
**good paper, but still needs some clarifications**

**Rating:** 6
**Confidence:** 3

**Review:**

##########################################################################
Summary:

The manuscript proposes a unified model which combines label propagation algorithm (LPA) and graph convolution network (GCN). The main idea is to optimize edge weights (after making edge weights trainable) by maximizing the intra-class feature influence. Introducing the theorems on the relationship between feature and label influence, and LPA’s prediction, the authors propose the unified objective function (a summation of the GCN loss and LPA loss) which combines both methods.
##########################################################################
Reasons for score:

Overall, I vote for accepting. The manuscript is overall well written, and the motivation of the proposed method is well explained by the proposed theorems. The logic leading to the objective function sounds reasonable and interesting. However, the manuscript still seems to need some clarifications.

##########################################################################Pros:

1.	The manuscript provides a new theoretical viewpoint to combine LPA and GNN.
2.	The proposed method outperforms the baselines, including some state-of-the-art Methods.

##########################################################################
Cons:

1. The method is a transductive method (test data points should are present during training) although the authors briefly mentioned the possibility of extension to inductive learning. I think that additional specialized effort is needed to extend label propagation in inductive setting. I think this is one of main weakness of the model.
2. Regarding the term regularization term for the LPA loss. This is generally referring to a penalty term on a penalty on the model complexity. I think that it is more appropriate if the authors introduced the proposed objective function as a simple implementation of multi-objective optimization.

#########################################################################
Questions:
In the experimental results. “In addition, we propose another baseline GCN+LPA, with simply adds predictions of”: It is not clear how to combine the prediction solutions from both methods (adding predictions in different scales?).

For Figure 5. What is the sparseness of edges in the random graphs? GCN-LPA has additional many parameters (the size is equal to the number of non-zero edges in the graph) compared to GCN. I expected that the time complexity of GCN-LPA would be way larger than that of GCN. Is this experimental setting close to real prediction problems?

---

> ### Author Response · Authors · 2020-11-13
> **Authors' Response to Reviewer 3**
>
> We thank the reviewer for helpful and detailed feedback. The reviewer made a number of helpful suggestions, and we have addressed your comments and included further clarifications in the paper to make the paper clearer and more understandable.
>
> Q1: *Additional specialized effort is needed to extend label propagation in inductive setting.*
>
> A1: The proposed method can be generalized to inductive setting as long as the edge weights is not modeled using its identity but a function of features of its two end-points. Please refer to the last paragraph in page 4 for details.
>
>
> Q2: *I think that it is more appropriate if the authors introduced the proposed objective function as a simple implementation of multi-objective optimization.*
>
> A2: We agree with the reviewer that it is a great intuition and explanation for the proposed loss term. We have added this part in the paper.
>
>
> Q3: *How is GCN+LPA implemented?*
>
> A3: GCN+LPA is implemented by simply averaging the predicted probability vectors output by GCN and LPA. These two vectors are within the same scale since they are both normalized.
>
>
> Q4: *What is the sparseness of edges in random graphs in Figure 5?*
>
> A4: The average number of neighbors for each node in random graph is set as 5 (this is clarified under Figure 6). This setting is reasonable since we can see from Table 3 that the average node degree in most real-world datasets is quite small ($\leq 5$).

---

> ### Author Response · Authors · 2020-11-24
> **Rebuttal Follow-up**
>
> Dear reviewer,
>
> Thanks again for your time reading our paper and rebuttal. we have addressed your questions and concerns in the rebuttal and we would really appreciate your feedback. Please let us know if there is any other information we can provide to assist in evaluating our paper. Thanks very much!

---

### Official Review · AnonReviewer1 · 2020-10-29
**need some improvement**

**Rating:** 5
**Confidence:** 4

**Review:**

This paper addresses the problem that edges in a graph could be noisy, containing erroneous edges. With the assumption of GCN that ‘labels/features are correlated over the edges of the graph’, it is desired that weights of inter-class edges are large, and those of intra-class edges are small. Hence, these noisy edges could impair GCN’s performance.

Addressing this problem, this work propose to optimize the given adjacency matrix based on the performance of Label Propagation(LPA) on it. LPA also assume the ‘homophily’ property of graphs, hence adapting adjacency matrix on it can encourage larger weights being given to trajectories linking two same-class nodes. This step is expected to be beneficial for the performance of GCN.

This paper has following strong points:
1.	Proposed GCN-LPA can learn to optimize the edges and perform node classification in an end-to-end manner, without causing much more training time cost;
2.	The writing is clear and easy to understand. The motivation, theory part is presented step by step, with complete proofs of those theorem;
3.	Experiments show that the proposed model is better at splitting embedding of nodes from different classes, and achieve improvement in node classification performance.
Following are the weak points:
1.	The idea that LPA can help GCN is not convincing. These two methods have similar assumption over the data, as also shown in the relation between label influence weight and feature influence weight in Theorem 1. Hence, why the authors claim on Page4, after Eq(9), that only updating edges with GCN will cause overfitting? Would not they be the same?
2.	The experiment is not very complete. Implemented GCN-LPA optimizes edges with gradient from both LPA and GCN tasks. The idea that LPA contains complementary information for GCN is not well justified. I would suggest that using LPA to obtain the updated edges and then fix them, before sending them to the GCN model.
3.	Besides, there are some other papers also seeking to adapt edges for the training of GCN, which are not mentioned. For example,
[Jiang, Bo, et al. "Semi-supervised learning with graph learning-convolutional networks." Proceedings of the IEEE Conference on Computer Vision and Pattern Recognition. 2019.]
[Yang, Liang, et al. "Topology Optimization based Graph Convolutional Network." IJCAI. 2019.]

---

> ### Author Response · Authors · 2020-11-13
> **Authors' Response to Reviewer 1**
>
> We thank the reviewer for helpful and detailed feedback. The reviewer made a number of helpful suggestions, and we have addressed your comments and included further clarifications in the paper to make the paper clearer and more understandable.
>
> Q1: *Why the authors claim that updating edges only with GCN will cause overfitting?*
>
> A1: When we say that “updating edges only with GCN will cause overfitting”, we are comparing a (normal) GCN whose learnable parameters are transformation matrix W, with another GCN whose learnable parameters are transformation matrix W and adjacency matrix A. Simultaneously learning W and A in GCN will certainly increase the risk of overfitting since the number of learnable parameters increases. The reason why introducing an LPA loss term can alleviate this problem is that LPA loss term provides **direct** supervision for learning edge weights.
>
>
> Q2: *I would suggest that using LPA to obtain the updated edges and then fix them, before sending them to the GCN model.*
>
> A2: We agree with the reviewer that this is an alternative to combine GCN and LPA. We have tried this method but it does not perform as well as the current version.
>
>
> Q3: *Missing related work.*
>
> A3: Thanks very much for the information. We have added these work in our paper.

---

> ### Author Response · Authors · 2020-11-24
> **Rebuttal follow-up**
>
> Dear reviewer,
>
> Thanks again for your time reading our paper and rebuttal. we have addressed your questions and concerns in the rebuttal and we would really appreciate your feedback. Please let us know if there is any other information we can provide to assist in evaluating our paper. Thanks very much!

---

### Official Review · AnonReviewer4 · 2020-10-30
**Official Blind Review #4**

**Rating:** 3
**Confidence:** 4

**Review:**

This paper aims to combine the label propagation and graph convolutional network with the modeling of their latent relationships. In the developed model, the node label is utilized to infer the edge weights between different nodes. From the evaluation results in Section 4, the performance improvement between the proposed method GCN-LPA and GDC is marginal, which can hardly demonstrate the advantage of the unified model (with GCN and LPA) over the graph diffusion network (without the restriction of information aggregation over neighboring nodes).

Furthermore, this work generates another baseline with the combination of prediction GCN and LPA methods. From the evaluation results, this baseline performs much worse than GCN and GAT, which may indicate that the predict combination involves some noise. It is better to conduct further experiments to show the effectiveness of the proposed GCN and LPA integration mechanism over simplified combination.

A minor note would be the lack of detailed hyperparameter tuning strategies of compared baselines. Different parameter settings may offer different performances; thus, it would be better to report how to perform the parameter tuning over various compared methods (such as GAT, GCN and GDC), to achieve good model performance and ensure a fair performance comparison.

The proposed method incorporates the label propagation to calculate edge weights, which share similar paradigm for learning node correlations with graph attention network and its extensions. In addition to the performance gap between the GCN-LPA and GAT, more clarifications about the model difference could be added, to have a better understanding of the new combined GCN and LPA framework.

It would be better to show the performance as the training/test ratio varies. It will be interesting to see is more data helpful to capture graph structural information better.

In the experiments, only model scalability comparison between the new GCN-LPA method and GCN, is studied. Is the GCN-LPA more efficient than other baselines, and which component of the new framework is more computationally expensive?

---

> ### Author Response · Authors · 2020-11-13
> **Authors' Response to Reviewer 4**
>
> We thank the reviewer for helpful and detailed feedback. The reviewer made a number of helpful suggestions, and we have addressed your comments and included further clarifications in the paper to make the paper clearer and more understandable.
>
> Q1: *The performance improvement between the proposed method and GDC is marginal, which can hardly demonstrate the advantage of the unified model over the graph diffusion network.*
>
> A1: Our method outperforms GDC in most cases, which demonstrates its advantage over GDC. It is true that the performance gain is not statistically significant, but given the large number of SOTA GNNs proposed in recent two years, it is extremely hard to achieve significant performance gain over every baseline on every dataset. Even GDC itself does not significantly outperform every baseline on every dataset according to their reported results. We think that the result in this paper is sufficient to demonstrate the effectiveness of our method. Besides, our contribution is more than the empirical effectiveness but also the theoretical analysis on GCN and LPA.
>
>
> Q2: *The result of the baseline GCN+LPA is worse than GCN and LPA.*
>
> A2: The reason that GCN+LPA performs worse than GCN is straightforward: it simply averages the predicted results of GCN and LPA, but the performance of LPA is quite bad. Therefore, LPA is a drag on the performance of GCN+LPA. The purpose of setting this baseline is to show that simply averaging their results does not work well. In fact, we did have designed another baseline called LPA->GCN, which uses LPA to produce pseudo-labels for unlabeled nodes then training on GCN using all labels (including the pseudo-labels). The result of this baseline is slightly better than GCN+LPA but still much worse than our proposed method. We do not show its result in this paper because we think that LPA+GCN is already sufficient to show that simply combining these two modules does not work well.
>
>
> Q3: *The lack of detailed hyperparameter tuning strategies of compared baseline.*
>
> A3: The hyperparameter settings are the same as reported in their original papers or codes. This is fair because we are using the same datasets and the same experimental settings (e.g. train/test ratio) as baselines.
>
>
> Q4: *What is the model difference between the proposed method and GAT?*
>
> A4: This is clarified in Abstract (the No. 5 line from the bottom: “*Our model can also be seen as…*”), Introduction (the No. 6 line from the bottom of the second last paragraph: “*It is worth noticing that…*”), and Related Work (the last paragraph: “*Attention and Diffusion on Graphs*...”). Basically, the difference between the proposed method and GAT is that, in GAT, edge weights are learned based on **node feature similarity**, while our work learns edge weights based on **node labels**.
>
>
> Q5: *It would be better to show the performance as the training/test ratio varies.*
>
> A5: This is shown in Table 2, where we vary the labeled node rate from 5% to 80%. A large labeled node rate will increase our performance gain because our method relies on node labels to reweight edges.
>
>
> Q6. *Only model scalability comparison between GCN-LPA method and GCN is studied.*
>
> A6: We only compare the running time between GCN-LPA and GCN because their implementations are quite similar, except that GCN-LPA introduces an extra LPA regularizer. Therefore, comparing their running time is reasonable. We do not compare with other baselines because the running time largely depends on their implementations (programming framework, number of epochs, early stopping, number of hidden dimensions...) The comparison will be reasonable only if they are implemented under the same framework. This is a huge work that we may not able to complete within the rebuttal period, but we will add the comparison between GCN-LPA and other baselines in the future version.

---

> ### Author Response · Authors · 2020-11-24
> **Rebuttal follow-up**
>
> Dear reviewer,
>
> Thanks again for your time reading our paper and rebuttal. we have addressed your questions and concerns in the rebuttal and we would really appreciate your feedback. Please let us know if there is any other information we can provide to assist in evaluating our paper. Thanks very much!

---

### Official Review · AnonReviewer2 · 2020-11-07
**concerns about the experiments**

**Rating:** 5
**Confidence:** 4

**Review:**

The paper combines the label propagation algorithm (LPA) and graph convolutional neural networks (GCNs) and proposes a unified model with learnable edge weights utilizing both feature and label influence.
The theoretical analysis of the correlation between LPA and GCN is interesting. Based on the theoretic analysis they show that the key to improving the performance of GCN is to enable nodes of the same class to connect more strongly, thus they propose to use LPA to reweight the edges and then use the new edge weights for GCN. However, their final model jointly learns edge weights with both LPA and GCN, which is acceptable but might reduce the connection to the theory.
Moreover, my main concerns are about the experiments.
(1)	Some of the baselines do not match the results reported in the original paper (e.g. GAT has much better performance than the numbers reported in Table 1); and the improvement of accuracy is actually marginal on most datasets. It will be better if the authors can analyze in which condition or on what types of datasets GCN-LPA works better.
(2)	I would like to see an ablation study which removes the LPA loss (\lambda L_{lpa}(A)) or replaces it with l2 regularization in Equation (9). Since the main idea of this paper is to reweight the edges with LPA, it is necessary to show the effect of the LPA regularization term. Reweighting the edges using only GCN is a very natural ablation model.

---

> ### Author Response · Authors · 2020-11-13
> **Authors' response to Reviewer 2**
>
> We thank the reviewer for helpful and detailed feedback. The reviewer made a number of helpful suggestions, and we have addressed your comments and included further clarifications in the paper to make the paper clearer and more understandable.
>
> Q1: *Some of the baselines do not match the results reported in the original paper (e.g. GAT has much better performance than the numbers reported in Table 1).*
>
> A1: As shown in the literature (e.g., Table 1 in [Shchur et al.](https://arxiv.org/pdf/1811.05868.pdf) and Figure 3 in [Klicpera et al.](https://proceedings.neurips.cc/paper/2019/file/23c894276a2c5a16470e6a31f4618d73-Paper.pdf)), GAT does not outperform other methods by a large margin (actually GAT does not achieve the SOTA performance on most datasets).
>
>
> Q2: *It will be better if the authors can analyze in which condition or on what types of datasets GCN-LPA works better.*
>
> A2: Since our method relies on labels to reweight edges, it works better on datasets that: 1) exhibit strong homophily, and 2) have sufficient number of labeled nodes.
>
>
> Q3: *I would like to see an ablation study which removes the LPA loss ($\lambda L_{lpa}(A)$) or replaces it with l2 regularization in Equation (9).*
>
> A3: This is already shown in Figure 4, where $\lambda$ measures the weight of LPA loss term in the loss function in Eq (9). When $\lambda=0$, the LPA loss term is removed.

---

> ### Author Response · Authors · 2020-11-24
> **Rebuttal follow-up**
>
> Dear reviewer,
>
> Thanks again for your time reading our paper and rebuttal. we have addressed your questions and concerns in the rebuttal and we would really appreciate your feedback. Please let us know if there is any other information we can provide to assist in evaluating our paper. Thanks very much!

---

### Decision · Program_Chairs · 2021-01-07
**Final Decision**

**Decision:**

Reject

**Comment:**

Three of the reviewers are significantly concerned about this submission while R3 was positive during review. During discussion, R3 also agreed that there are concerns not only on experimental designs and results but also the proposed model. Thus a reject is recommended.